# Information Retrieval and Awareness about Evidence-Based Dentistry among Dental Undergraduate Students—A Comparative Study between Students from Malaysia and Finland

**DOI:** 10.3390/dj8030103

**Published:** 2020-09-03

**Authors:** Pentti Nieminen, Eswara Uma, Sudipta Pal, Marja-Liisa Laitala, Olli-Pekka Lappalainen, Eby Varghese

**Affiliations:** 1Medical Informatics and Data Analysis Research Group, University of Oulu, 90014 Oulu, Finland; 2Faculty of Dentistry, Melaka Manipal Medical College, Manipal Academy of Higher Education (MAHE), 75150 Melaka, Malaysia; eswara.uma@manipal.edu.my (E.U.); sudipta_paul@yahoo.com (S.P.); eby.varghese@manipal.edu.my (E.V.); 3Research Unit of Oral Health Sciences, Faculty of Medicine, University of Oulu, 90014 Oulu, Finland; marja-liisa.laitala@oulu.fi; 4Faculty of Medicine, University of Helsinki, 00014 Helsinki, Finland; olli-pekka.lappalainen@helsinki.fi

**Keywords:** clinical appraisal, evidence-based dentistry, dental students, literature retrieval, Malaysia, Finland

## Abstract

Background: A fundamental skill in education includes the ability to search for, evaluate, and synthesize information, and this cannot be underestimated in dental education. The aim of this study was to assess how dental students from Malaysia and Finland acquire scientific information and to compare their information retrieval skills. Methods: Fourth and fifth-year dental students from Malaysia and Finland were invited to participate. A self-administered structured questionnaire including items about the use of information sources, subjective assessment of literature retrieval skills and knowledge was used. Results: A total of 226 dental students participated in the survey: 131 from Malaysia and 95 from Finland. In both countries, the highest interest for data retrieval among students was found in the oral surgery specialty. The three most used sources of information among Malaysian students were personal lecture notes, dental textbooks, and colleagues; while Finnish students used colleagues, lecture notes, and current clinical guidelines. Students’ knowledge of evidence-based practice was inadequate in both student groups. Though the majority of participants reported that they had good or passable skills in literature retrieval, more students from Finland judged themselves to have at least good skills compared to those from Malaysia. Conclusion: Dental education in both countries includes information retrieval studies and mandatory research projects. However, students did not often use those sources that are considered essential in evidence-based dentistry. Universities should further develop educational and training interventions that guide students to use knowledge resources more effectively for critically appraising scientific evidence.

## 1. Introduction

In the present era of easy access to online resources for both patients and doctors, clinicians are expected to be up to date with the latest treatment modalities, dental materials, and research being conducted all over the world. The abundance of online resources makes it difficult for clinicians to evaluate the validity, reliability, and quality of the information provided [1]. This confusion has led to the concept of “evidence-based medicine” (EBM). In 1996, EBM was defined as the judicious integration of the best scientific clinical evidence and clinical expertise in order to arrive at a clinical decision regarding a patient keeping in mind their personal choices and preferences [2]. The main objective was to strengthen the scientific basis of medicine in order to clarify any uncertainties regarding the treatment plan [3]. Like any other specialty, the same principle was used in the field of dentistry to combine the best scientific evidence and clinical expertise with patient outcomes. Evidence-based dentistry (EBD) integrates clinical experience, the best available research evidence, and patient values and expectations [4]. In order to produce such competent clinicians, dental schools all over the world are emphasizing EBD in their curriculum and providing opportunities for students to become competent in using EBD to guide them in patient care [5]. Studies have shown that to practice EBD and to make a correct clinical decisions, a dentist must follow the five steps of forming an answerable question: collect the best available evidence from electronic databases; critically appraise the evidence for its validity, reliability, relevance, and bias; integrate the evidence with one’s own clinical expertise and the patient’s needs and preferences; and evaluate the EBD process and the final result [6,7,8,9].

Different types of research work can help in retrieving evidence. Primary research includes original work that can be in the form of cross-sectional surveys, longitudinal cohort studies, randomized controlled trials, case-control studies, or case reports [8]. Secondary research work includes systematic reviews and meta-analyses of a cluster of similar primary research, critical appraisals, books, or clinical practice guidelines as well as information available on webpages [8].

Due to the large number of published articles in numerous databases and also the complexity of critically appraising such evidence, the search for high-quality evidence in dentistry is becoming increasingly difficult [10,11,12]. Studies carried out in different parts of the world have revealed a lack of knowledge about evidence-based dental practice, but the majority have shown a positive attitude in adopting it in the future [13,14]. Poor knowledge, low awareness, and lack of time and resources are some of the perceived barriers among practicing dentists [13,15,16]. As a result, undergraduate education of dental students plays an important part in teaching the principles of EBD to promote lifelong learning [17,18].

The dental curriculum in Malaysia has a duration of five years. The first two years are preclinical years and the next three are clinical. In the second year of study, the students conduct a mentor-structured project, and at the end they submit a report of the project conducted. Students in Malaysia are introduced to the concept of evidence-based dentistry and research methodology through a mandatory research project as well as through clinico-pathological-case presentations (CPCP). In CPCP, the students are given a clinical topic. Students need to conduct a relevant literature search and then present a case report related to the clinical topic. The curriculum mandates that students undertake a research project under the guidance of a faculty member for a period of two years. During this period, the students first undergo twelve weeks of research methodology classes. Under the guidance of the respective faculty supervisors, students prepare a proposal, obtain ethical approval, collect data, conduct statistical analysis, and prepare a report, and these are later presented at a student conference. In order to carry out their research project, the students must search and critically appraise the available evidence. Both exercises help students in their day-to-day clinical work, as the students have mandatory clinics from the third to fifth years of study where they do clinical procedures related to all branches of dentistry.

The dental curriculum in Finland is five years. The undergraduate curriculum includes a course (Knowledge Management and Research) on scientific thinking and the principles of scientific research. In addition, 10–20 weeks are reserved for advanced special studies in dental research. These studies are compulsory and include an independent research project and the writing of a short master thesis. Students are encouraged to consider research projects of any kind, but most select their topic from ongoing scientific research projects in the university’s departments and clinics. The academic staff at the departments act as supervisors for the thesis [19]. The dental curriculum also includes clinical patient work in institutional training clinics during the third to fifth years of study. However, the students are also allowed to practice in public health care during their vacations after four years of study. For a student in Finland, one important criterion for becoming a general dental practitioner is the ability to acquire information from several databases and use this information effectively in a scientific and critical manner to treat their patients [20]. The curriculum, teaching and learning approaches, or strategies may be different between Asian and European countries and may have an influence on the practice of seeking evidence-based information. There are also significant cultural difference which may affect these processes. This comparative study between similar cohorts of students from two different countries from two continents will help to assess undergraduate dental curricula and provide areas for improvement if required.

While both groups of students in the two different countries need to be well acquainted with EBD, this study aims to compare the information retrieval and awareness regarding EBD among both groups. The main aims of this study are as follows:to investigate in which dental specialties students search for scientific information and knowledge and to trace differences between student groups from Malaysia and Finland;to examine how dental students from the two different countries acquire scientific information as a dental student and during training at dental clinics;to compare the level of awareness of dental students regarding evidence-based dentistry between dental schools and year of study;to assess how students evaluated their skills in literature retrieval by dental school and year of study.

## 2. Materials and Methods 

This cross-sectional survey was conducted to compare the knowledge of EBD and literature retrieval skills among dental students from Malaysia and Finland. The study subjects comprised fourth- and fifth-year students from one Malaysian and one Finnish dental institution. The fourth- and fifth-year students were asked to answer an anonymous questionnaire during an appropriate lecture or demonstration at the end of 2018. The sample size was limited to the total number of 235 registered fourth and fifth-year undergraduate dental students in these institutions in 2018.

The instrument used was a questionnaire modified from a previous study among Finnish dental students [20]. It was slightly edited to better adapt to our international study with participants from different countries using various retrieval methods and databases. The self-administered questionnaire included questions about gender, dental specialties related to their recent research, their use of information sources during the past 6 months (at the university campus area, at home, or at work), and their self-assessment of literature retrieval skills and knowledge of EBD. Students did not receive benefits or credits for participating in the survey. The updated version of the questionnaire is included as the Appendix A.

The total number of registered undergraduate (fourth and fifth year) dental students in the Malaysian dental institution at the time was 145, and the survey attained a high response rate of 90% (131/145). The response rate in Finland was 93% with 95 of the 102 registered students in the Finnish dental institution completing the questionnaire. 

Presuming that 40% of the student population had good literature retrieval skills [20] and a nonresponse rate of 10%, a sample size of 145 students from Malaysia and 102 students from Finland (assuming allocation ratio n_1_/n_2_ = 1.4 between the two groups) was calculated to be large enough. This allowed at least 90% power to correctly detect a statistically significant (α = 0.05) difference of 20% in skills between the two student groups.

Students from both institutions gave consent to participate. The study protocol was approved by the Research and Ethics Committee of the Malaysian dental institution (MMMC/FOD/AR/E C-2018, 12th of September 2018). According to the guidelines of the Ministry of Education and Culture in Finland, survey studies with anonymous questionnaires do not need an approval from an ethics committee.

To assess the topics (dental specialties) of their most recent searches, the students had to answer the question: ‘For which dental specialties have you been looking for scientific information from different information sources? Select three specialties about which you most frequently searched for information’ with the following options: oral and maxillofacial surgery, oral pathology, endodontics, prosthodontics, periodontology/parodontology, cariology, orthodontics, oral radiology, implantology, pedodontics/pediatric dentistry, other.

A set of questions was focused on the use of information sources: (a) “How often have you used the following information sources as an undergraduate student in the last six months?”; (b) “How often have you used the following information sources at the polyclinics in the college (Malaysia) or at the dental clinic (Finland) during the past 6 months?” 

We also evaluated the students’ literature retrieval skills using a scale from 1 (excellent) to 5 (poor). Their answers to this question were grouped into three categories: inadequate (1–2), passable (3), and good (4–5). In addition, the questionnaire also included three questions that assessed the students’ knowledge of EBD. Some students had difficulty understanding a question assessing the level of evidence, and their interpretation of the question was not the same as was intended by the designer of the questionnaire. This question was not included in the analyses. The analyzed multiple-choice questions tested knowledge of the terms meta-analysis and PICO.

### Data Analysis

The frequency and percentage distributions of student characteristics (gender, year of study, and perceived literature retrieval skills) were reported for participants from Malaysia and Finland. Percentage distribution was used to estimate the proportion of students who searched for scientific information within different dental specialties. The use of different information sources and levels of awareness regarding evidence-based dentistry were the main outcome variables. Their frequency and percentage distributions were presented by the year of study and dental school. A chi-square test was used to evaluate the statistical significance of differences in the frequency tables. We used IBM SPSS Statistics 25 for data analysis. The data that support the findings of this study are available from the corresponding author upon reasonable request. 

## 3. Results

Our sample included a total of 226 dental students from Malaysia and Finland (Table 1). The majority of the subjects (65.3%) were female, with an almost equal number of students from the fourth and fifth years of study from both dental schools. Students were asked about which dental specialties they had searched for scientific information and knowledge (Table 2). The majority of students (67.9% from Malaysia and 69.5% from Finland) reported searching topics related to oral and maxillofacial surgery. More than half of the Malaysian students sought information on oral medicine, oral pathology, and endodontics. Students from Finland reported searching for information on cariology and prosthodontics more often than Malaysian students. Table 3 reports the percentage of students using various sources of information as a dental student and during training at a dental clinic. An overwhelming majority of the students from Finland (98%) reported asking their colleagues for help while studying at the university campus. However, this was seen only among 74.8% of Malaysian students. Personal lecture notes were also frequently used (about 92%) among both student groups. When studying in general, students from Malaysia used dental textbooks (77.3%) more often than students from Finland (36.8%). 

While training at the dental clinic, the students from Finland most frequently sought information from colleagues (95.8%), current clinical guidelines (61.5%), and personal lecture notes (54.7%). Malaysian students most often used their personal lecture notes (86.2%), dental textbooks (82.4%), and colleagues (78.6%).

We also administered an EBD test to the students (Table 4). A much higher percentage of Finnish fourth-year students were able to identify the correct answer to the question “What is meta-analysis?” compared to Malaysian students (78.7% vs. 26.1%, *p*-value of chi-square test <0.001). However, a slightly higher percentage of Malaysian fifth-year students reported to have better knowledge than Finnish fifth year students about meta-analyses (67.7% vs. 60.4%, *p* = 0.007). The differences in knowledge on EBD between fourth- and fifth-year students within the countries were also statistically significant: in Malaysia, fifth-year students knew about meta-analyses more than their younger colleagues (*p* <0.001), but in Finland the fourth-year students performed better (*p* = 0.007).

Dental students in Malaysia and Finland demonstrated poor knowledge of evidence-based practice (Table 4). Only 14 (6.2%) of 226 students answered the question about PICO correctly. More Malaysian students were able to identify the correct meaning of the abbreviation PICO. The knowledge of this evidence-based term was poor both among Fourth and Fifth-year students (Table 4).

Respondents were also asked to self-evaluate their literature retrieval skills. Figure 1 shows the distribution of the students’ perceived skills in information retrieval by dental school and year of study. A total of 66 (29.2%) evaluated their skills in literature retrieval as good or excellent. Whereas the majority of participants reported that they had good or passable skills, more students from Finland than from Malaysia judged themselves to have at least good skills in literature retrieval (fourth-year: 57.4% vs. 11.6%, fifth-year: 43.8% vs. 16.1%) (*p* <0.001 for both years of the study groups). There were no significant differences in self-evaluated skills between fourth- and fifth-year students from Malaysia (*p* = 0.634). In Finland, the younger students evaluated their skills as good (57.4%) and passable (36.2%), while the fifth-year students were more critical (good 43.8% and passable 56.3%). These differences were statistically significant (*p* = 0.036). A total of 35.9% of students from Malaysia reported having only mediocre or poor skills while this proportion was only 3.2% among dental students from Finland.

## 4. Discussion

Application of the best evidence from the literature is becoming increasingly important in dentistry. The present study investigated how dental students from Malaysia and Finland acquire scientific information and how familiar they were with methods for different information sources. In both countries, students reported searching for knowledge most often in oral and maxillofacial surgery. Among Malaysian students, the three most commonly used sources of information were personal lecture notes, dental textbooks, and colleagues, while Finnish students used colleagues, lecture notes, and current clinical guidelines. The Malaysian and Finnish dental students recognized themselves to be quite unfamiliar with EBD terminology, but a high proportion of students had knowledge about meta-analysis. Interestingly, a higher proportion of Malaysian than Finnish students reported having only mediocre or poor literature retrieval skills 

In searching for scientific information and knowledge in different dental specialties, there were variations between dental schools. However, differences between the curricula explain this finding. Malaysian students searched for information on subjects that were going to be assessed in the exit exam of that year as well as subjects that were part of their competency assessment. In Finland, cariology is an independent discipline, taught together with endodontics and pedodontics, whilst in Malaysia, various aspects of cariology are taught in oral pathology, conservative dentistry, pediatric dentistry, and community dentistry. However, in both countries, oral and maxillofacial surgery (OMFS) seems to be the specialty bringing the highest interest for data retrieval among students. This finding is in line with a previous study from Finland [20]. This may indicate that the discipline encourages students to actively utilize information resources.

Malaysian students reported to use textbooks and lecture notes as they are informed by their faculty. The faculty share lecture notes that have the current guidelines incorporated in them, and students further compile their own personal lecture notes. This could possibly explain their greater dependency on personal lecture notes, more than their Finnish counterparts, and their greater focus on current guidelines. In a situation where students cannot access their personal notes, students still can access any information while in the clinics, including their class notes, by using computers that are kept in a designated room. Further, after college hours, the students can access information through computers in the library, and most students do use them for their research work. The library also arranges article retrievals for students related to their academic or research work if they are unable to get online. In Finland, the students seem to lean strongly on current clinical guidelines (https://www.kaypahoito.fi/en/). These guidelines are the “Gold Standard” in Finland, and they are built on evidence-based data and include updated recent scientific literature. University lecturers also share this same information in lecture notes, and the students consider them as a reliable source of information. In the training clinic, a personal computer is available in every single treatment unit, and all web-based data sources are exploitable. Therefore, textbooks or lecture notes are not popular among students. In both countries, the ability to consult colleagues seemed to be important and in Finland, this is encouraged by teachers. 

Our results suggest that dental students did not primarily search and read scientific papers, which is a key feature for practicing evidence-based dentistry. This finding is consistent with studies from other medical sub-fields [21,22,23]. It is possible that students have difficulty reading scientific papers, and this can be attributed to several factors, including unfamiliar terminology or research techniques in scientific papers, difficulty understanding statistics, having to wade through numerous papers to find the information, and inadequate training to identify the clinical application of knowledge gained from scientific journal articles [22,24]. Earlier studies have also stated that dental students lack resources for retrieving evidence-based information, which is a barrier for moving into this direction [13,14,15]. However, they maintained a positive attitude and expressed a desire to incorporate it in the future.

Malaysian students were able to respond better due to their involvement in their research projects, where they have to use the literature extensively. While they were using the PICO format to formulate their research questions, they were not sure of the full form of PICO. Fifth-year students did better than their Fourth-year colleagues, as the former complete their research and compile their report with the help of a literature search to substantiate their findings. It is concerning that Finnish dental students did not know the PICO principles, though their meta-analysis skills were reported to be good. This might be due to the early course allocation in the first and second preclinical years of the curriculum, and the students may already have forgotten these skills during the study years. It is, therefore, important to emphasize the role of PICO throughout the course. However, our survey did not focus on assessing EBD practices among students.

Malaysian students’ self-perception of their information retrieval skills was reported to be mediocre or poor. These students conduct a clinical or nonclinical research project for which they have to review the relevant recent literature. After going through the process of conducting research, they probably perceive their ability to retrieve information as inadequate, owing to their inability to use the right keywords or databases to obtain the required information. Hence, they need to be guided by their respective supervisors on how to retrieve the right information. They could also be too self-critical and under-rate themselves despite being good. In Finland, students reported their retrieval skills to be at least passable or even good. The reason for this is most likely due to the students’ strong orientation to retrieve data by themselves utilizing databases throughout their studies. The Finnish students’ responses may have focused on retrieving information to solve clinical and practical issues rather than searching scientific evidence. Similar to the Malaysian dental education system, all students in Finland also participate in a scientific research project during their course and complete a master’s thesis [20]. A course on literature retrieval, scientific writing, and communication, as well as data analysis take place in the first and second years of study, which may reflect on the results of this survey. 

The curriculum in Malaysia is spread over five years. Presently it is subject-based, and most schools have moved, or are moving towards an integrated curriculum. The first two years are preclinical years, while the last three years are clinical years where the students start treating patients while on rotation in various specialties. Introduction to the concept of research occurs in the preclinical years, where the students conduct a mentor-structured project (MSP). In the clinical years, full elective research is conducted that starts in year 4 and is completed in year 5. This early introduction through MSP helps the students in their information retrieval and data analysis. Additionally, the students seek information to prepare for seminars and case-based learning sessions.

The dental education curriculum in Finland is 5.5 years long, with two preclinical years and three and a half clinical years, which is quite similar to the Malaysian curriculum. The last six months of study in Finland is entirely clinical training completed in municipal dental care centers, and it is based on normal daily dental practice but under the guidance of senior consultant practitioner. The structure of the curriculum includes plenty of cross-disciplinary studies. The teaching of data retrieval and scientific writing is not fully comparable to the Malaysian curriculum with regard to duration and contents. A course on knowledge management and research is mandatory in the Finnish dental curriculum as part of the preclinical studies, i.e., at the beginning of the first year of study. The course includes an introduction to scientific writing (structure of assignments and thesis, reporting of findings, tables and figures, citing and references), literature retrieval (basic literature retrieval methods and use of bibliographic databases available at the university), data analysis (basic statistical methods and use of statistical software), and scientific communication (scientific journals, research articles, critical evaluation of research findings, ethics in scientific publication and bibliometrics). The purpose of introducing this mandatory course in scientific research and literature retrieval at the very beginning of the curriculum is to provide these basic skills before conducting more subject-oriented dental studies. This has helped dental students to learn, practice, and apply methods for critically appraising scientific articles and evidence in their further studies. 

The participation rates in both countries were excellent. One limitation of this study should be noted: this was performed in local settings in one dental institution in Malaysia and in one dental institution in Finland, and each educational setting is unique. Nevertheless, in spite of the limited scope, our findings might be helpful when considering possible educational and training interventions in dental schools that teach, practice, and apply methods for critically appraising scientific evidence.

## 5. Conclusions

Our multi-institutional study on dental education in two culturally different countries provides useful insights into the strengths and weaknesses of final-year undergraduate students in information retrieval and awareness regarding EBD. We conclude that students from both countries did not primarily search for and read scientific papers. However, students acquired scientific information differently in Finland than in Malaysia. Among the Finnish students, the sources for information were colleagues, lecture notes, and clinical guidelines. Malaysian students referred to lecture notes, textbooks, and colleagues for obtaining information. The levels of awareness for dental students regarding evidence-based dentistry and meta-analysis could be improved in both dental schools. The self-reported literature retrieval skill was passable to good among Finnish students; however, it was mediocre or poor among Malaysian students. It is noteworthy that dental education in both countries includes information retrieval studies and mandatory research projects. However, students did not often use those sources, which are considered essential in evidence-based medicine, e.g., Cochrane Library, Medline, scientific journals. The most recent findings published in the scientific literature should be the primary source for literature reviews and the critical evaluation of treatment methods. Universities have the responsibility to teach their dental students and dentists various knowledge management methods to cope with scientific information. Dental students be more knowledgeable than their patients are and provide evidence to support their answers to patients’ questions.

## Figures and Tables

**Figure 1 dentistry-08-00103-f001:**
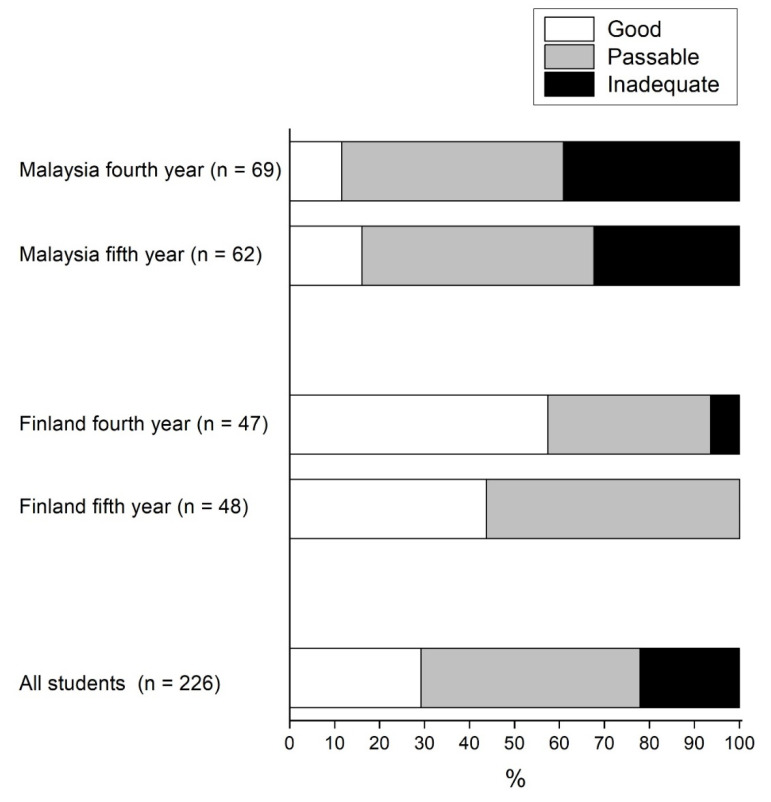
The percentage distributions of dental students’ perceived skills in information retrieval by dental school and year of study.

**Table 1 dentistry-08-00103-t001:** The frequency and percentage distributions of gender, year of study, and perceived skills of dental students from Malaysia (*n* = 131) and Finland (*n* = 95).

	Malaysia	Finland	All
	*n* (%)	*n* (%)	*n* (%)
Gender			
Male	41 (31.8)	36 (38.7)	77 (34.7)
Female	88 (68.2)	57 (61.3)	145 (65.3)
Year of study			
Fourth	69 (52.7)	47 (49.5)	116 (51.3)
Fifth	62 (47.3)	48 (50.5)	110 (48.7)

Gender: Four missing values.

**Table 2 dentistry-08-00103-t002:** Percentages of most recent searches according to dental specialty and school (*n* = 226).

Specialty Searched	Dental School	
	Malaysia (*n* = 131)	Finland (*n* = 95)
Oral and maxillofacial surgery	67.9	69.5
Oral medicine and oral pathology	64.9	33.8
Endodontics	59.5	36.8
Prosthodontics	22.9	43.2
Periodontology/Parodontology	18.3	33.7
Cariology	6.1	44.2
Orthodontics	14.5	31.6
Oral radiology	23.7	14.7
Implantology	11.5	2.1
Pedodontics/Pediatric dentistry	7.6	2.1
Other/cosmetic, aesthetic, community	2.3	0

**Table 3 dentistry-08-00103-t003:** Daily or weekly use of information sources as an undergraduate student and when training at the dental clinic among fourth and fifth-year students from Malaysia (*n* = 131) and Finland (*n* = 95).

Source of Information	When Studying in General				When Training at the Dental Clinic			
	Malaysia		Finland		Malaysia		Finland	
	Fourth Year (%) (*n* = 69)	Fifth Year (%) (*n* = 62)	Fourth Year (%) (*n* = 47)	Fifth Year (%) (*n* = 48)	Fourth Year (%) (*n* = 69)	Fifth Year (%) (*n* = 62)	Fourth Year (%) (*n* = 47)	Fifth Year (%) (*n* = 48)
Colleagues	76.8	72.6	97.9	97.9	81.2	75.8	97.9	93.8
Personal lecture notes	94.2	88.7	91.5	93.8	91.3	80.6	53.2	56.3
Current clinical guidelines	20.3	9.7	63.8	79.2	17.4	21.0	53.2	68.8
Other textbooks	30.4	14.5	43.5	37.5	39.1	9.7	8.5	6.3
Dental textbooks	82.6	59.7	21.3	52.1	92.8	71.0	8.5	10.4
Medline/PubMed	24.6	21.0	14.9	18.8	18.8	19.4	2.1	2.1
National dental journals	2.9	4.8	10.6	10.4	4.3	8.1	0	2.1
Other sources	1.4	4.8	0	2.1	0	0	0	2.1
Other dental journals	1.4	3.2	0	6.3	4.3	6.5	0	2.1
Other guidelines	0	0	0	2.1	0	0	0	2.1
Advertisements	18.8	29.0	o	8.3	13.0	16.1	0	2.1
EBM journals	4.3	4.8	0	4.2	2.9	8.3	0	2.1
Cochrane library	5.8	6.5	2.1	8.3	5.8	3.2	0	4.2

**Table 4 dentistry-08-00103-t004:** The frequency and percentage distributions of dental students’ knowledge of EBD concepts among fourth- and fifth-year students from Malaysia (*n* = 131) and Finland (*n* = 95).

Knowledge about EBD Concepts	Fourth Year		Fifth Year	
	Malaysia (%)	Finland (%)	Malaysia (%)	Finland (%)
**What is meta-analysis?**				
A statistical method in which results from several studies have been combined and analyzed using quantitative methods	18 (26.1)	37 (78.7)	42 (67.7)	29 (60.4)
Tool to evaluate the quality of systematic reviews	2 (2.9)	0	9 (14.5)	10 (20.8)
Summary of studies with clear clinical significance	0	8 (17.0)	0	14 (14.7)
I don’t know	49 (71.0)	2 (4.3)	11 (17.7)	5 (5.3)
**What is PICO?**				
Abbreviation to describe the elements of good clinical questions	2 (2.9)	0	8 (12.9)	4 (8.3)
Abbreviation of a collaboration group, which has defined criteria for treatment guidelines	2 (2.9)	3 (6.7)	8 (12.9)	5 (10.4)
Abbreviation for evidence-based treatment phases in dental clinic	9 (13.2)	0	9 (14.5)	1 (2.1)
I don’t know	55 (80.9)	42 (93.3)	37 (59.7)	38 (79.2)

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
