# Peer review of "Information Retrieval and Awareness about Evidence-Based Dentistry among Dental Undergraduate Students—A Comparative Study between Students from Malaysia and Finland"

_dentistry, 2020, doi:10.3390/dj8030103_

Round 1

Reviewer 1 Report

The methodology of the manuscript it is valid and well proposed. The sample is well represented and selected. The English language is often distracting some corrections need to improve it.

INTRODUCTION

This paragraph introduces the research and produce a well-done short literature review about the topic. In this section is adequately presented the sample of undergraduate students involved in the study. However, it would be necessary to end the paragraph with the reasons that led to develop this study and finally to reassess the aim of the study with precision.

The MATERIALS AND METHODS section presents some lacks:

- please add the significance level selected for the study.

- clarify if an error evaluation was made during the measurements.

- please define at the beginning of the paragraph the characteristics of the study.

- please clarify if the approval of the Ethics Committee has been obtained for this research and possibly report the protocol number.

- please define the calculation of the sample size.

RESULTS

This paragraph is well developed and bring out the data resulting from the measurements.

DISCUSSION

the discussion section is well developed even though I think it is slightly verbose and extensive. I, therefore, suggest summarizing some concepts and removing some parts that could be redundant in the text.

CONCLUSIONS

Not all the conclusions reported response to the aim of the study. Some of the sentences are not supported by results.

REFERENCES

The references are right for the developed topic, but I would suggest only adding this further:

Rodriguez y Baena R, Lupi SM, Pastorino R, Maiorana C, Lucchese A, Rizzo S.Radiographic evaluation of regenerated bone following poly(Lactic-Co-Glycolic) acid/hydroxyapatite and deproteinized bovine bone graft in sinus lifting. J Craniofac Surg 2013; May 24 (3): 845-48

TABLES

They are clear and help clarify the data that emerged from the study and support both the results section and the discussions section.

Reviewer 2 Report

The knowledge about evidence-based dentistry (EBD) and the mode of knowledge acquisition were investigated using a questionnaire administered to dental students from two different countries.  Respondents were sizeable - 95 students from Finland and 131 from Malaysia. The introduction section is brief but sufficient to provide a background to the study. Standard procedure was followed for data collection. The methodology is simple and clearly explained. Large part of this study is descriptive. However, part of the data sets was analysed using Chi-square tests. Results are adequately presented providing a comparative narrative of the two cohorts of students. The differences in the knowledge and self-perception of the two cohorts regarding EBD have been adequately discussed with respect to dental curricula of the two countries.

It is desirable that a copy of the questionnaire is made available for publication as an additional material to the paper.

The manuscript should be closely checked for typographical errors and clarity of expressions. Some minor points listed below should be addressed by the authors:

  1. Please correct the spelling of the word ‘retrieval’ in the title.
  2. Page 2, line 57: Please delete word ‘on’ to read as ‘emphasizing EBD’.
  3. Page 2, Table 2: Please change ‘Oral Surgery’ to ‘Oral and Maxillofacial Surgery’.
  4. Page 5, line 171: The figures quoted as “78.9% vs 22.1%” do not match with those given in Table 4. These should be “78.7% vs 26.1%”.
  5. Page 6, line 188: Please change ’57,4’ to ‘57.4’.
  6. Page 7, lines 210-211: The following sentence is not clear. Please revise it: “However, differences between … the results”.

Reviewer 3 Report

                The manuscript by Nieminen et al reports the results of surveying dental students in Finland and Malaysia regarding evidence-based dentistry (EBD) knowledge and practices.  This is an important topic that impacts clinical practice and a dedication to life-long learning.  Some of the results from the surveys were interesting but it is not clear that the study design was adequate to meet the objective of the study.

There are several points for consideration:

  1. “Retrieval” is misspelled in the title.

  1. The Abstract concludes that “Students’ knowledge of evidence-based practice was inadequate in both student groups.” This statement is based solely on a survey response that queried whether the students were familiar with the acronym ‘PICO’.  It is conceivable, perhaps even likely, that a significant number of students were familiar with the principles of PICO but not the acronym.

  1. The end of the Abstract is contradictory. It states that the universities provided opportunities to students to learn about evidence-based dentistry and then two sentences later states that this is something the universities should do as if it were missing in the curriculum.

  1. How much autonomy do dental students have in drafting a treatment plan? It seems an evaluation of EBD should examine dentists in practice and how they choose to keep up with the literature and new developments.

  1. The manuscript does not provide any explanation for why dental schools in Finland and Malaysia were chosen to participate.

  1. It is not clear that the survey questions truly assess EBD knowledge and practice.

Round 2

Reviewer 1 Report

Considering the changes made I believe that the work is ready for publication

Author Response

1. Moderate English changes required.

Response 1: The manuscript has now been carefully revised by an English editing service provided by MDPI.

2. Considering the changes made I believe that the work is ready for publication

Response 2: Thank you.

Reviewer 3 Report

The authors have been moderately responsive to the initial critiques and the changes have improved the manuscript.  There are some limitations to the experimental design that cannot be remedied.  There are numerous minor instances where grammatical improvements are necessary.

Author Response

1. The authors have been moderately responsive to the initial critiques and the changes have improved the manuscript.  There are some limitations to the experimental design that cannot be remedied.  There are numerous minor instances where grammatical improvements are necessary.

Response 1: Thank you for the understanding comment about the study design. The manuscript has now been carefully revised by an English editing service provided by MDPI.